# Minibatch and Momentum Model-based Methods for Stochastic Weakly Convex Optimization

**Qi Deng**[1]          **Wenzhi Gao**[2]

School of Information Management and Engineering
Shanghai University of Finance and Economics

[1]qideng@sufe.edu.cn   [2]gwz@163.shufe.edu.cn

## Abstract

Stochastic model-based methods have received increasing attention lately due to their appealing robustness to the stepsize selection and provable efficiency guarantee. We make two important extensions for improving model-based methods on stochastic weakly convex optimization. First, we propose new minibatch model-based methods by involving a set of samples to approximate the model function in each iteration. For the first time, we show that stochastic algorithms achieve linear speedup over the batch size even for non-smooth and non-convex (particularly, weakly convex) problems. To this end, we develop a novel sensitivity analysis of the proximal mapping involved in each algorithm iteration. Our analysis appears to be of independent interests in more general settings. Second, motivated by the success of momentum stochastic gradient descent, we propose a new stochastic extrapolated model-based method, greatly extending the classic Polyak momentum technique to a wider class of stochastic algorithms for weakly convex optimization. The rate of convergence to some natural stationarity condition is established over a fairly flexible range of extrapolation terms.

While mainly focusing on weakly convex optimization, we also extend our work to convex optimization. We apply the minibatch and extrapolated model-based methods to stochastic convex optimization, for which we provide a new complexity bound and promising linear speedup in batch size. Moreover, an accelerated model-based method based on Nesterov's momentum is presented, for which we establish an optimal complexity bound for reaching optimality.

## 1   Introduction

In this paper, we are interested in the following stochastic optimization problem:

$$\min_{x \in \mathcal{X}} \quad f(x) = \mathbb{E}_{\xi \sim \Xi}\big[f(x, \xi)\big] \tag{1}$$

where $f(\cdot, \xi)$ stands for the loss function, sample $\xi$ follows certain distribution $\Xi$, and $\mathcal{X}$ is a closed convex set. We assume that $f(\cdot, \xi)$ is weakly convex, namely, the sum of $f(x, \xi)$ and a quadratic function $\frac{\lambda}{2}\|x\|^2$ is convex ($\lambda > 0$). This type of non-smooth non-convex functions can be found in a variety of machine learning applications, such as phase retrieval, robust PCA and low rank decomposition [8]. To solve problem (1), we consider the stochastic model-based method (SMOD, [14, 9, 1]), which comprises a large class of stochastic algorithms (including stochastic (sub)gradient descent, proximal point, among others). Recent work [14, 9] show that SMOD exhibits promising convergence property: both asymptotic convergence and rates of convergence to certain stationarity

35th Conference on Neural Information Processing Systems (NeurIPS 2021).

measure have been established for the SMOD family. In addition, empirical results [9, 15] indicate that SMOD exhibits remarkable robustness to hyper-parameter tuning and often outperforms SGD.

Despite much recent progress, our understanding of model-based methods for weakly convex optimization is still quite limited. Particularly, it is still unknown whether SMOD is competitive against modern SGD used in practice. We highlight some important remaining questions. First, despite the appealing robustness and stable convergence, the SMOD family is sequential in nature. It is unclear whether minibatching, which is immensely used in training learning models, can improve the performance of SMOD when the problem is non-smooth. Particularly, the current best complexity bound $\mathcal{O}(\frac{L^2}{\varepsilon^4})$ from [9], which is regardless of batch size, is unsatisfactory. Were this bound tight, a sequential algorithm (using one sample per iteration) would be optimal: it offers the highest processing speed per iteration as well as the best iteration complexity. Therefore, it is crucial to know whether minibatching can improve the complexity bound of the SMOD family or the current bound is tight. Second, in modern applications, momentum technique has been playing a vital role in large-scale non-convex optimization (see [31, 28]). In spite of its effectiveness, to the best of our knowledge, momentum technique has been provably efficient only in **1)** unconstrained smooth optimization [23, 10, 19] and **2)** non-smooth optimization with a simple constraint [25], which constitute only a portion of the interesting applications. From the practical aspect, it is peculiarly desirable to know whether momentum technique is applicable beyond in SGD and whether it can benefit the SMOD algorithm family in the non-smooth and non-convex setting.

**Contributions.** Our work is motivated by the aforementioned challenge to make SMOD more practically efficient. We summarize the contributions as follows. First, we extend SMOD to the minibatch setting and develop sharper rates of convergence to stationarity. Leveraging the tool of algorithm stability ([6, 27, 20]), we provide a nearly complete recipe on when minibatching would be helpful even in presence of non-smoothness. Our theory implies that stochastic proximal point and stochastic prox-linear are inherently parallelizable: both algorithms achieve linear speedup over the minibatch size. To the best of our knowledge, this is the first time that these minibatch stochastic algorithms are proven to exhibit such an acceleration even for *non-smooth* and *non-convex* (particularly, *weakly convex*) optimization. Moreover, our theory recovers the complexity of minibatch (proximal) SGD in [9], showing that (proximal) SGD enjoys the same linear speedup by minibatching for smooth composite problems with non-smooth regularizers or with constrained domain.

Second, we present new extrapolated model-based methods by incorporating a Polyak-type momentum term. We develop a unified Lyapunov analysis to show that a worst-case complexity of $\mathcal{O}(1/\varepsilon^4)$ holds for all momentum SMOD algorithms. To the best of our knowledge, these are the first complexity results of momentum stochastic prox-linear and stochastic proximal point for non-smooth non-convex optimization. Since our analysis offers complexity guarantees for momentum SGD and its proximal variant, our work appears to be more general than a recent study [25], which only proves the convergence of momentum projected SGD. Proximal SGD is more advantageous in composite optimization, where the non-smooth term is often involved via its proximal operator rather than the subgradient. For example, in the Lasso problem, it is often favorable to invoke the proximal operator of $\ell_1$ function (Soft-Thresholding) to enhance solution sparsity. We summarize the complexity results in Table 1.

Third, we develop new convergence results of SMOD for convex optimization, showing that minibatch extrapolated SMOD achieves a promising linear speedup over the batch size under some mild condition. Specifically, to obtain some $\varepsilon$-optimal solution, our proposed method exhibits an $\mathcal{O}(1/\varepsilon + 1/(m\varepsilon^2))$ complexity bound in the worst case. Moreover, we develop a new minibatch SMOD based on Nesterov's momentum, achieving the $\mathcal{O}(1/\varepsilon^{1/2} + 1/(m\varepsilon^2))$ optimal complexity bound. Note that a similar complexity result, explicitly relying on the smoothness assumption, has been shown in a recent study [7]. Compared to this work, our analysis makes weaker assumptions, showing that smoothness is not a must-have for many model-based algorithms, such as SPL and SPP, to get sharper complexity bound.

**Other related work.** For smooth and composite optimization, it is well known that SGD can be linearly accelerated by minibatching (c.f. [11, 18, 29]). Minibatch model-based methods have been studied primarily in the convex setting. Asi et al. [2] investigates the speedups of minibatch stochastic model-based methods in the convex smooth, restricted strongly convex and convex interpolation settings, respectively. Since their assumptions differ from ours, the technique does not readily apply to the non-convex setting. Chadha et al. [7] studies the accelerated minibatch model-based methods

Table 1: Complexity of `SMOD` to reach $\mathbb{E}\,\|\nabla_{1/\rho}f\| \leq \varepsilon$ (M: minibatch; E: Extrapolation, $m$: batch size)

| Algorithms | Problem | Current Best | Ours |
|---|---|---|---|
| M + SGD | $f$: non-smooth | $\mathcal{O}(1/\varepsilon^4)$[9] | $\mathcal{O}(1/\varepsilon^4)$ |
| M + Prox. SGD | $f = \ell + \omega$; $\ell$:smooth | $\mathcal{O}(1/(m\varepsilon^4) + 1/\varepsilon^2)$[9] | $\mathcal{O}(1/(m\varepsilon^4) + 1/\varepsilon^2)$ |
| M + SPL/SPP | $f$: non-smooth | $\mathcal{O}(1/\varepsilon^4)$[9] | $\mathcal{O}(1/(m\varepsilon^4) + 1/\varepsilon^2)$ |
| E + SGD | $f$: non-smooth | $\mathcal{O}(1/\varepsilon^4)$[25] | $\mathcal{O}(1/\varepsilon^4)$ |
| E + Prox. SGD | $f = \ell + \omega$; $\ell$:smooth | — | $\mathcal{O}(1/\varepsilon^4)$ |
| E + SPL/SPP | $f$: non-smooth | — | $\mathcal{O}(1/\varepsilon^4)$ |
| M + E + SGD | $f$: non-smooth | $\mathcal{O}(1/\varepsilon^4)$[25] | $\mathcal{O}(1/\varepsilon^4)$ |
| M + E + Prox. SGD | $f = \ell + \omega$; $\ell$:smooth | — | $\mathcal{O}(1/(m\varepsilon^4) + 1/\varepsilon^2)$ |
| M + E + SPL/SPP | $f$: non-smooth | — | $\mathcal{O}(1/(m\varepsilon^4) + 1/\varepsilon^2)$ |

for convex smooth and convex interpolated problems. The interpolation setting, where the model can perfectly fit the data, is not considered in our paper. Algorithm stability [6, 27]—an important technique for analyzing the generalization performance of stochastic algorithms [20, 3], is the key tool to obtain some of our convergence results. In contrast to the traditional work, our paper employs the stability argument to obtain sharper optimization convergence rates (with respect to the batch size). See Section 3. As noted by an anonymous reviewer, a similar idea of using stability analysis was proposed by Wang et al. [30], albeit with a different motivation from distributed stochastic optimization. Robustness and fast convergence of model-based methods have been shown on various statistical learning problems [8, 15, 1, 4, 16, 5]. Drusvyatskiy and Paquette [13] give a complete complexity analysis of the accelerated proximal-linear methods for deterministic optimization. Zhang and Xiao [32] further improve the convergence rates of prox-linear methods on certain finite-sum and stochastic problems by using variance-reduction. Momentum and accelerated methods for convex stochastic optimization can be referred from [24, 26]. The study [10, 23, 31] develop the convergence rate of stochastic momentum method for smooth non-convex optimization.

## 2   Background

Throughout the paper, we use $\|\cdot\|$ to denote the Euclidean norm and $\langle\cdot,\cdot\rangle$ to denote the Euclidean inner product. We assume that $f(x)$ is bounded below. i.e., $\min_x f(x) > -\infty$. The subdifferential $\partial f(x)$ of function $f(x)$ is the set of vectors $v \in \mathbb{R}^d$ that satisfy: $f(y) \geq f(x) + \langle v, y - x\rangle + o(\|x - y\|)$, as $y \to x$. Any such vector in $\partial f(x)$ is called a subgradient and is denoted by $f'(x) \in \partial f(x)$ for simplicity. We say that a point $x$ is stationary if $0 \in \partial f(x) + N_{\mathcal{X}}(x)$, where the normal cone $N_{\mathcal{X}}(x)$ is defined as $N_{\mathcal{X}}(x) \triangleq \{d : \langle d, y - x\rangle \leq 0, \forall y \in \mathcal{X}\}$. For a set $S$, define the set distance to $0$ by: $\|\mathcal{S}\|_{-} \triangleq \inf\{\|x - 0\|, x \in \mathcal{S}\}$. It is natural to use the quantity $\|\partial f(x) + N_{\mathcal{X}}(x)\|_{-}$ to measure the stationarity of point $x$.

**Moreau-envelope.** The $\mu$-Moreau-envelope of $f$ is defined by $f_\mu(x) \triangleq \min_{y \in \mathcal{X}} \left\{ f(y) + \frac{1}{2\mu}\|x - y\|^2 \right\}$ and the proximal mapping associated with $f(\cdot)$ is defined by $\mathrm{prox}_{\mu f}(x) \triangleq \mathrm{argmin}_{y \in \mathcal{X}} \left\{ f(y) + \frac{1}{2\mu}\|x - y\|^2 \right\}$. Assume that $f(x)$ is $\lambda$-weakly convex, then for $\mu < \lambda^{-1}$, the Moreau envelope $f_\mu(\cdot)$ is differentiable and its gradient is $\nabla f_\mu(x) = \mu^{-1}(x - \mathrm{prox}_{\mu f}(x))$.

The `SMOD` family iteratively computes the proximal map associated with a model function $f_{x^k}(\cdot, \xi_k)$:

$$x^{k+1} = \underset{x \in \mathcal{X}}{\mathrm{argmin}} \left\{ f_{x^k}(x, \xi_k) + \frac{\gamma_k}{2}\|x - x^k\|^2 \right\}, \tag{2}$$

where $\{\xi_k\}$ are i.i.d. samples. Typical algorithms and the accompanied models are described below.

**Stochastic (Proximal) Gradient Descent**: consider the composite function $f(x, \xi) = \ell(x, \xi) + \omega(x)$ where $\ell(x, \xi)$ is a data-driven and weakly-convex loss term and $\omega(x)$ is a convex regularizer such as $\ell_1$-penalty. `SGD` applies the model function:

$$f_y(x, \xi) = \ell(y, \xi) + \left\langle \ell'(y, \xi), x - y\right\rangle + \omega(x). \tag{3}$$

**Stochastic Prox-linear** (`SPL`): consider the composition function $f(x, \xi) = h(C(x, \xi))$ where $h(\cdot, \xi)$ is convex continuous and $C(x, \xi)$ is a continuously differentiable map. We perform partial

linearization to obtain the model

$$f_y(x,\xi) = h\big(C(y,\xi) + \langle \nabla C(y,\xi), x - y\rangle\big). \tag{4}$$

**Stochastic Proximal Point (**SPP**)**: compute (2) with full stochastic function:

$$f_y(x,\xi) = f(x,\xi). \tag{5}$$

Throughout the paper, we assume that $f(x,\xi)$ is continuous and $\mu$-weakly convex, and that the model function $f_x(\cdot,\cdot)$ satisfies the following assumptions [9].

**A1:** For any $\xi \sim \Xi$, the model function $f_x(y,\xi)$ is $\lambda$-weakly convex in $y$ ($\lambda \geq 0$).

**A2:** Tightness condition: $f_x(x,\xi) = f(x,\xi)$, $\forall x \in \mathcal{X}$, $\xi \sim \Xi$.

**A3:** One-sided quadratic approximation: $f_x(y,\xi) - f(y,\xi) \leq \frac{\tau}{2}\|x - y\|^2$, $\forall x, y \in \mathcal{X}, \xi \sim \Xi$.

**A4:** Lipschitz continuity: There exists $L > 0$ that $f_x(z,\xi) - f_x(y,\xi) \leq L\|z - y\|$, for any $x, y, z \in \mathcal{X}$, $\xi \sim \Xi$.

*Remark* 1. Assumption A2 is quite standard and will be used only in the convergence proof. Combining A1 and A3, we immediately have that $f(x,\xi)$ is $(\lambda + \tau)$-weakly convex. Thus, it suffices to assume that $\mu < \tau + \lambda$. Assumptions A2-A4 can be slightly relaxed by replacing the uniform bound with a bound on expectation over $\xi$, leading to only a minor adjustment to the analysis.

Denote $\hat{x} \triangleq \operatorname{prox}_{f/\rho}(x) = \operatorname{argmin}_y \left\{ f(y) + \frac{\rho}{2}\|y - x\|^2 \right\}$ for some $\rho > \mu$. Davis and Drusvyatskiy [9] revealed a striking feature of Moreau envelope to characterize stationarity:

$$\|\hat{x} - x\| = \rho^{-1}\|\nabla f_{1/\rho}(x)\|, \text{ and } \|\partial f(\hat{x}) + N_{\mathcal{X}}(\hat{x})\|_- \leq \|\nabla f_{1/\rho}(x)\|.$$

Namely, a point $x$ with small gradient norm $\|\nabla f_{1/\rho}(x)\|$ stays in the proximity of a nearly-stationary point $\hat{x}$. With this observation, they show the first complexity result of SMOD for non-smooth non-convex optimization: $\min_{1 \leq k \leq K} \mathbb{E}[\|\nabla f_{1/\rho}(x^k)\|]^2 \leq \mathcal{O}(\frac{L}{\sqrt{K}})$. Note that this rate is regardless of the size of minibatches since it does not explicitly use any information of the samples other than the Lispchitzness of the model function. Due to this limitation, it remains unclear whether minibatching can further improve the convergence rate of SMOD.

## 3   SMOD **with minibatches**

In this section, we present a minibatch SMOD method which takes a small batch of i.i.d. samples to estimate the model function. The overall procedure is detailed in Algorithm 1. Within each iteration, Algorithm 1 forms a stochastic model function $f_{x^k}(\cdot, B_k) = \frac{1}{m_k}\sum_{i=1}^{m_k} f_{x^k}(x, \xi_{k,i})$ parameterized at $x^k$ by sampling over $m_k$ i.i.d. samples $B_k = \xi_{k,1}, \ldots, \xi_{k,m_k}$. Then it performs proximal update to get the next iterate $x^{k+1}$. We will illustrate the main convergence results of Algorithm 1 and leave all the proof details in Appendix sections. But first, let us present an additional assumption.

**A5:** Two-sided quadratic bound: for any $x, y \in \mathcal{X}, \xi \sim \Xi$, $\left| f_x(y,\xi) - f(y,\xi) \right| \leq \frac{\tau}{2}\|x - y\|^2$.

*Remark* 2. Assumption A5 is vital for our improved convergence analysis. While it is slightly stronger than A3, A5 is indeed satisfied by the SMOD family in most contexts: **1)** For SPP, A5 is trivially satisfied by taking $f_x(y,\xi) = f(y,\xi)$. **2)** For SPL, we minimize a composition function $f(x,\xi) = h(C_\xi(x))$ where $h(\cdot)$ is a $c_1$-Lipschitz convex function and $C_\xi(\cdot)$ is a $c_2$-Lipschitz smooth map. In view of (4), A5 is verified with $|f_x(y,\xi) - f(y,\xi)| \leq c_1\left\|C_\xi(y) - C_\xi(x) - \nabla C_\xi(x)^{\mathrm{T}}(y - x)\right\| \leq \frac{c_1 c_2}{2}\|x - y\|^2$. **3)** For SGD, A5 is satisfied if $\ell(\cdot,\xi)$ is $c_3$-Lipschitz smooth for some $c_3 > 0$, as $|f_x(y,\xi) - f(y,\xi)| \leq |\ell(y,\xi) - \ell(x,\xi) - \nabla\ell(x,\xi)^{\mathrm{T}}(y - x)| \leq \frac{c_3}{2}\|x - y\|^2$. We note that A5 is not satisfied by SGD when the loss $\ell(\cdot,\xi)$ is also non-smooth. Unfortunately, there seems to be little hope to accelerate SGD in such a case since the convergence rate of SGD already matches the rate of deterministic subgradient method.

We present an improved complexity analysis of SMOD by leveraging the framework of algorithm stability [6, 27]. In stark contrast to its standard application in characterizing the algorithm generalization performance, stability analysis is applied to determine how the variation of a minibatch affects the *estimation of the model function* in each algorithm iteration.

---

**Algorithm 1** Stochastic Model-based Method with Minibatches (`SMOD`)

---

**Input:** $x^1, \gamma_k$;
**for** $k = 1$ **to** $K$ **do**
    Sample a minibatch $B_k = \{\xi_{k,1}, \ldots, \xi_{k,m_k}\}$ and update $x^{k+1}$ by solving

$$\min_{x \in \mathcal{X}} \left\{ \frac{1}{m_k} \sum_{i=1}^{m_k} f_{x^k}(x, \xi_{k,i}) + \frac{\gamma_k}{2} \|x - x^k\|^2 \right\} \tag{6}$$

**end for**

---

**Notations.** Let $B = \{\xi_1, \xi_2, \ldots, \xi_m\}$ be a batch of i.i.d. samples and $B_{(i)} = B \setminus \{\xi_i\} \cup \{\xi_i'\}$ by replacing $\xi_i$ with an i.i.d. copy $\xi_i'$, and $B' = \{\xi_1', \xi_2', \ldots, \xi_m'\}$. Let $h(\cdot, \xi)$ be a stochastic model function, and denote $h(y, B) = \frac{1}{m} \sum_{i=1}^{m} h(y, \xi_i)$. The stochastic proximal mapping associated with $h(\cdot, B)$ is defined by $\mathrm{prox}_{\rho h}(x, B) \triangleq \mathrm{argmin}_{y \in \mathcal{X}} \left\{ h(y, B) + \frac{1}{2\rho} \|y - x\|^2 \right\}$ for some $\rho > 0$. We denote $x_B^+ \triangleq \mathrm{prox}_{\rho h}(x, B)$ for brevity. We say that the stochastic proximal mapping $\mathrm{prox}_{\rho h}$ is $\varepsilon$-stable if, for any $x \in \mathcal{X}$, we have

$$\left| \mathbb{E}_{B, B', i} \left[ h(x_{B_{(i)}}^+, \xi_i') - h(x_B^+, \xi_i') \right] \right| \leq \varepsilon, \tag{7}$$

where $i$ is an index chosen from $\{1, 2, \ldots, m\}$ uniformly at random.

The next lemma exploits the stability of proximal mapping associated with the model function.

**Lemma 3.1.** *Let $f_z(\cdot, B)$ be a stochastic model function under the assumptions A1-A4. For $\gamma \in (\lambda, \infty)$, vectors $z$ and $y$, the proximal mapping $\mathrm{prox}_{f_z/\gamma}(y, B) = \mathrm{argmin}_{x \in \mathcal{X}} \left\{ f_z(x, B) + \frac{\gamma}{2} \|x - y\|^2 \right\}$ is $\varepsilon$-stable with $\varepsilon = \frac{2L^2}{m(\gamma - \lambda)}$.*

Applying Lemma 3.1, we obtain the error bound for approximating the full model function in the next theorem.

**Theorem 3.2.** *Under all the assumptions of Lemma 3.1, we have*

$$\left| \mathbb{E}_{B_k} \left[ f_{x^k}(x^{k+1}, B_k) - \mathbb{E}_\xi f_{x^k}(x^{k+1}, \xi) | \sigma_k \right] \right| \leq \varepsilon_k, \ \varepsilon_k = \frac{2L^2}{m_k(\gamma_k - \lambda)}. \tag{8}$$

*where $\sigma_k$ is the $\sigma$-algebra generating $\{B_i\}_{1 \leq i \leq k-1}$.*

Note that since $x^{k+1}$ is dependent on $B_k$, $f_{x^k}(x^{k+1}, B_k)$ is not an unbiased estimator of $\mathbb{E}_\xi[f_{x^k}(x^{k+1}, \xi)]$. However, the stability argument identifies that the expected approximation error is a decreasing function of batch size $m_k$. This observation is the key to the sharp analysis of minibatch stochastic algorithms. With all the tools at our hands, we obtain the key descent property in the following theorem.

**Theorem 3.3.** *Suppose that $\rho > \lambda + \tau$, $\gamma_k \geq \rho + \tau$, A5 and all the assumptions in Lemma 3.1 hold. Let $\mathbb{E}_k[\cdot]$ abbreviates $\mathbb{E}_{B_k}[\cdot | \sigma_k]$ and $\varepsilon_k$ be given by (8), then we have*

$$\frac{(\rho - \lambda - \tau)}{\rho(\gamma_k + \rho - 2\lambda - \tau)} \|\nabla f_{1/\rho}(x^k)\|^2 \leq f_{1/\rho}(x^k) - \mathbb{E}_k[f_{1/\rho}(x^{k+1})] + \frac{\rho \varepsilon_k}{\gamma_k + \rho - 2\lambda - \tau}. \tag{9}$$

Next, we specify the rate of convergence to stationarity using a constant stepsize policy.

**Theorem 3.4.** *Under the assumptions of Theorem 3.3, let $\Delta = f_{1/\rho}(x^1) - \min_x f(x)$, $m_k = m$, and $\gamma_k = \gamma = \max\{\rho + \tau, \lambda + \eta\}$ where $\eta = \frac{\sqrt{K}}{\alpha_0 \sqrt{m}}$ and $\alpha_0 \in (0, \infty)$. Let $k^*$ be an index chosen in $\{1, 2, \ldots, K\}$ uniformly, then we have*

$$\mathbb{E}\left[ \|\nabla f_{1/\rho}(x^{k^*})\|^2 \right] \leq \frac{\rho}{\rho - \lambda - \tau} \left[ \frac{(2\rho - \lambda)\Delta}{K} + \left( \frac{\Delta}{\alpha_0} + 2\alpha_0 \rho L^2 \right) \frac{1}{\sqrt{mK}} \right]. \tag{10}$$

*Remark* 3. The performance of `SMOD` depends on $\alpha_0$ and batch size $m$. (10) implies that when batch size is fixed, the best rate is obtained at $\alpha_0^* = \sqrt{\frac{\Delta}{2\rho}} \frac{1}{L}$. Since both $\Delta$ and $L$ are unknown,

---

**Algorithm 2** Stochastic Extrapolated Model-Based Method (SEMOD)

---

**Input:** $x^0$, $x^1$, $\beta$, $\gamma$;
**for** $k = 1$ **to** $K$ **do**
  Sample data $\xi^k$ and update:

$$y^k = x^k + \beta(x^k - x^{k-1}) \tag{11}$$

$$x^{k+1} = \operatorname*{argmin}_{x \in \mathcal{X}} \left\{ f_{x^k}(x, \xi^k) + \frac{\gamma}{2}\|x - y^k\|^2 \right\} \tag{12}$$

**end for**

---

hyper-parameter tuning over $\alpha_0$ is required to obtain good empirical performance. For the simplicity of theoretical analysis, let us take $\alpha_0 = \alpha_0^*$. Hence, to obtain an iterate whose Moreau envelop has expected gradient norm smaller than $\varepsilon$, the total iteration count is $\mathcal{T}_\varepsilon = \max\left\{\mathcal{O}(\frac{\Delta}{\varepsilon^2}), \mathcal{O}(\frac{L^2\Delta}{m\varepsilon^4})\right\}$. For small batch size $m$ (i.e. $m = o(1/\varepsilon^2)$), the second term in $\max(,)$ dominates the bound $\mathcal{T}_\varepsilon$, yielding a total complexity of $\mathcal{O}(\frac{L^2\Delta}{m\varepsilon^4})$. Note that this complexity bound is better than the $\mathcal{O}(\frac{L^2\Delta}{\varepsilon^4})$ bound [9] by a factor of $m$.

*Remark* 4. Theorem 3.4 implies that SGD can be accelerated by minibatching on the smooth composite problems (3) but leaves out the more general problems where $\ell(x, \xi)$ is non-smooth and weakly convex. In the latter case, showing any improved rate of minibatch SGD is substantially more challenging. Without additional knowledge, the $\mathcal{O}(\frac{L^2\Delta}{\varepsilon^4})$ complexity of SGD already matches the best result for deterministic subgradient method (c.f. [9]). It remains unknown whether such $\mathcal{O}(1/\varepsilon^4)$ bound is tight or not, and a possible direction to obtain sharper complexity bound is by exploiting the non-smooth structure information such as sharpness.

**Solving the subproblems.** SGD is embarrassingly parallelizable by simply averaging the stochastic subgradients. We highlight how to solve the proximal subproblems for SPL and SPP. Consider the composition function $f(x, \xi) = h(C(x, \xi))$ where $h(a) = |a|$. For SPL, it is easy to transform the corresponding subproblem to an $\mathcal{O}(m_k)$-dimensional quadratic program (QP) in the dual space (e.g. [2]). The dual QP can be efficiently solved in parallel, for example, by a fast interior point solver. For SPP, we show that the subproblem can be solved by a deterministic prox-linear method at a rapid linear convergence rate. Note that the SPP subproblem is especially well-conditioned because our stepsize policy ensures a large strongly convex parameter $\gamma - \lambda$. We refer to the appendix for more technical details.

## 4 SMOD **with momentum**

We present a new model-based method by incorporating an additional extrapolation term, and we record this stochastic extrapolated model-based method in Algorithm 2. Each iteration of Algorithm 2 consists of two steps, first, an extrapolation step is performed to get an auxiliary update $y^k$. Then a random sample $\xi_k$ is collected and the proximal mapping, associated with the model function $f_{x^k}(\cdot, \xi_k)$, is computed at $y^k$ to obtain the new point $x^{k+1}$. For ease of exposition, we take constant values of stepsize and extrapolation term.

Note that Algorithm 2 can be interpreted as an extension of the momentum SGD by replacing the gradient descent step with a broader class of proximal mappings. To see this intuition, we combine (11) and (12) to get

$$x^{k+1} = \operatorname*{argmin}_{x \in \mathcal{X}} \left\{ f_{x^k}(x, \xi^k) + \gamma\beta\langle x^{k-1} - x^k, x - x^k\rangle + \frac{\gamma}{2}\|x - x^k\|^2 \right\}, \tag{13}$$

If we choose the linear model (3), i.e., $f_{x^k}(x, \xi^k) = f(x^k, \xi^k) + \langle f'(x^k, \xi^k), x - x^k\rangle$, and assume $\mathcal{X} = \mathbb{R}^d$, then the update (13) has the following form:

$$x^{k+1} = x^k - \gamma^{-1}f'(x^k, \xi^k) - \beta(x^{k-1} - x^k). \tag{14}$$

Define $v^k \triangleq \gamma(x^{k-1} - x^k)$ and apply it to (14), then Algorithm 2 reduces to the heavy-ball method

$$v^{k+1} = f'(x^k, \xi^k) + \beta v^k, \tag{15}$$

$$x^{k+1} = x^k - \gamma^{-1}v^{k+1}. \tag{16}$$

Despite such relation, the gradient averaging view (15) only applies to SGD for unconstrained optimization, which limits the use of standard analysis of heavy-ball method ([31]) for our problem. To overcome this issue, we present a unified convergence analysis which can deal with all the model functions and is amenable to both constrained and composite problems.

Our theoretical analysis of Algorithm 2 relies on a different potential function from the one in the previous section. Let us define the auxiliary variable

$$z^k \triangleq x^k + \frac{\beta}{1-\beta}(x^k - x^{k-1}). \tag{17}$$

The following lemma proves some approximate descent property by adopting the potential function $f_{1/\rho}(z^k) + \frac{\rho(\gamma\beta+\rho\beta^2\theta^{-2})}{2(\gamma\theta-\lambda\theta)}\|x^k - x^{k-1}\|^2$ and measuring the quantity of $\|\nabla f_{1/\rho}(z^k)\|$.

**Lemma 4.1.** *Assume that $\rho \geq 2(\tau + \lambda)$ and $\beta \in [0, 1)$. Let $\theta = 1 - \beta$. Then we have*

$$\frac{(\rho - \lambda\theta)}{2\rho(\gamma\theta - \lambda\theta)}\|\nabla f_{1/\rho}(z^k)\|^2 \leq f_{1/\rho}(z^k) - \mathbb{E}_k\left[f_{1/\rho}(z^{k+1})\right] + \frac{\rho L^2}{(\gamma\theta^2 - \rho\beta^2\theta^{-1})(\gamma\theta^2 - \lambda\theta^2)}$$

$$+ \frac{\rho(\gamma\beta + \rho\beta^2\theta^{-2})}{2(\gamma\theta - \lambda\theta)}\left(\|x^k - x^{k-1}\|^2 - \mathbb{E}_k[\|x^{k+1} - x^k\|^2]\right)$$

$$- \frac{\rho(\gamma - \rho\beta^2\theta^{-3})}{4(\gamma - \lambda)}\mathbb{E}_k[\|x^{k+1} - x^k\|^2]. \tag{18}$$

Invoking Lemma 4.1 and specifying the stepsize policy, we obtain the main convergence result of Algorithm 2 in the following theorem.

**Theorem 4.2.** *Under assumptions of Lemma 4.1, if we choose $x^1 = x^0$, and set $\gamma = \gamma_0\theta^{-1}\sqrt{K} + \lambda + \rho\beta^2\theta^{-3}$ for some $\gamma_0 > 0$, then*

$$\mathbb{E}[\|\nabla f_{1/\rho}(z^{k^*})\|^2] \leq \frac{2\rho}{\rho - \lambda}\left[\frac{\rho\beta^2\theta^{-2}\Delta}{K} + \left(\gamma_0\Delta + \frac{\rho L^2}{\theta\gamma_0}\right)\frac{1}{\sqrt{K}}\right] \tag{19}$$

*where $k^*$ is an index chosen in $\{1, 2, \ldots, K\}$ uniformly at random.*

*Remark* 5. Despite the fact that convergence is established for all $\gamma_0 > 0$, we can see that the optimal $\gamma_0$ would be $\gamma_0 = \sqrt{\frac{\rho}{\Delta\theta}}L$, which gives the bound $\mathbb{E}[\|\nabla f_{1/\rho}(z^{k^*})\|^2] \leq \frac{2\rho}{\rho - \lambda}\left(\frac{\rho\beta^2\theta^{-2}\Delta}{K} + 2L\sqrt{\frac{\rho\Delta}{\theta K}}\right)$. In practice, we can set $\gamma_0$ to a suboptimal value and obtain a possibly loose upper-bound.

*Remark* 6. Since $z^k$ is an extrapolated solution, it may not be feasible. It is desirable to show optimality guarantee at iterates $x^k$. Note that using Lemma 4.1 and the parameters in Theorem 4.2, it is easy to show that $\mathbb{E}[\|x^{k^*} - x^{k^*-1}\|^2] = \mathcal{O}(\frac{1}{K})$. Based on (17) we have $\|z^{k^*} - x^{k^*}\|^2 = \beta^2\theta^{-2}\mathbb{E}[\|x^{k^*} - x^{k^*-1}\|^2] = \mathcal{O}(\frac{1}{K})$. Using Lipschitz smoothness of Moreau envelop, we can show $\mathbb{E}[\|\nabla f_{1/\rho}(x^{k^*})\|^2]$ converges at the same $\mathcal{O}(\frac{1}{\sqrt{K}})$ rate as is shown in Theorem 4.2.

Combining momentum and minibatching, we develop a minibatch version of Algorithm 2 that takes a batch of samples $B_k$ in each iteration. The convergence analysis of this minibatch SEMOD is more involving. We leave the details in the Appendix but informally state the main result below.

**Theorem 4.3** (Informal)**.** *In the minibatch SEMOD, suppose that A5 holds, the batch size $|B_k| = m$ and $\gamma = \mathcal{O}(\sqrt{\frac{K}{m}})$, then $\mathbb{E}[\|\nabla f_{1/\rho}(z^{k^*})\|^2] = \mathcal{O}(\frac{1}{K} + \sqrt{\frac{1}{mK}})$.*

## 5   SMOD **for convex optimization**

Besides the study on non-convex optimization, we also apply model-based methods to stochastic convex optimization. Due to the space limit, we highlight main theoretical results but defer all the technical details to the Appendix section. We show that if certain assumption adapted from A5 for the convex setting holds, the function gap of minibatching SEMOD will converge at a rate of $\mathcal{O}\left(\frac{1}{K} + \frac{1}{\sqrt{mK}}\right)$. In view of this result, the deterministic part of our rate is consistent with the best $\mathcal{O}(\frac{1}{K})$ rate for the heavy-ball method. For example, see [12, 17]. Moreover, the stochastic part of the rate is improved from the $\mathcal{O}(\frac{1}{\sqrt{K}})$ rate of Theorem 4.4 [9] by a factor of $\sqrt{m}$.

An important question arises naturally: Can we further improve the convergence rate of model-based methods for stochastic convex optimization? Due to the widely known limitation of heavy-ball type momentum, it would be interesting to consider Nesterov's acceleration. To this end, we present a model-based method with Nesterov type momentum. Thanks to the stability argument, we obtain the following improved rate of convergence: $\mathcal{O}\left(\frac{1}{K^2} + \frac{1}{\sqrt{mK}}\right)$. We note that a similar convergence rate for minibatching model-based methods is obtained in a recent paper [7]. However, their result requires the assumption that the stochastic function is Lipschitz smooth while our assumption is much weaker.

## 6  Experiments

In this section, we examine the empirical performance of our proposed methods through experiments on the problem of robust phase retrieval. (Additional experiments on blind deconvolution are given in Appendix section). Given a set of vectors $a_i \in \mathbb{R}^d$ and nonnegative scalars $b_i \in \mathbb{R}_+$, the goal of phase retrieval is to recover the true signal $x^*$ from the measurement $b_i = |\langle a_i, x^* \rangle|^2$. Due to the potential corruption in the dataset, we consider the following penalized formulation

$$\min_{x \in \mathbb{R}^d} \quad \frac{1}{n} \sum_{i=1}^{n} \left| \langle a_i, x \rangle^2 - b_i \right| \tag{20}$$

where we impose $\ell_1$-loss to promote robustness and stability (cf. [15, 9, 25]).

**Data Preparation.** We conduct experiments on both synthetic and real datasets.

**1) Synthetic data.** Synthetic data is generated following the setup in [25]. We set $n = 300, d = 100$ and select $x^*$ from unit sphere uniformly at random. Moreover, we generate $A = QD$ where $Q \in \mathbb{R}^{n \times d}, q_{ij} \sim \mathcal{N}(0, 1)$ and $D \in \mathbb{R}^d$ is a diagonal matrix whose diagonal entries are evenly distributed in $[1/\kappa, 1]$. Here $\kappa \geq 1$ plays the role of condition number (large $\kappa$ makes problem hard). The measurements are generated by $b_i = \langle a_i, x^* \rangle^2 + \delta_i \zeta_i \ (1 \leq i \leq n)$ with $\zeta_i \sim \mathcal{N}(0, 25), \delta_i \sim$ Bernoulli$(p_{\text{fail}})$, where $p_{\text{fail}} \in [0, 1]$ controls the fraction of corrupted observations on expectation.

**2) Real data.** We consider `zipcode`, a dataset of $16 \times 16$ handwritten digits collected from [21]. Following the setup in [15], let $H \in \mathbb{R}^{256 \times 256}$ be a normalized Hadamard matrix such that $h_{ij} \in \left\{ \frac{1}{16}, -\frac{1}{16} \right\}, H = H^{\mathrm{T}}$ and $H = H^{-1}$. Then we generate $k = 3$ diagonal sign matrices $S_1, S_2, S_3$ such that each diagonal element of $S_k$ is uniformly sampled from $\{-1, 1\}$. Last we set $A = [HS_1, HS_2, HS_3]^{\mathrm{T}} \in \mathbb{R}^{(3 \times 256) \times 256}$. As for the true signal and measurements, each image is represented by a data matrix $X \in \mathbb{R}^{16 \times 16}$ and gets vectorized to $x^* = \text{vec}(X)$. To simulate the case of corruption, we set measurements $b = \phi_{p_{\text{fail}}}(Ax^*)$, where $\phi_{p_{\text{fail}}}(\cdot)$ denotes element-wise squaring and setting a fraction $p_{\text{fail}}$ of entries to 0 on expectation.

In the first experiment, we illustrate that `SMOD` methods enjoy linear speedup in the size of minibatches and exhibit strong robustness to the stepsize policy. We conduct comparison on `SPL` and `SGD` and describe the detailed experiment setup as follows.

**1) Dataset generation.** We generate four testing cases: the synthetic datasets with $(\kappa, p_{\text{fail}}) = (10, 0.2)$, and $(10, 0.3)$; `zipcode` with digit images of id 2 and 24;

**2) Initial point.** We set the initial point $x^1(= x^0) \sim \mathcal{N}(0, I_d)$ for synthetic data and $x^1 = x^* + \mathcal{N}(0, I_d)$ for `zipcode`;

**3) Stopping criterion.** We set the stopping criterion to be $f(x^k) \leq 1.5\hat{f}$, where $\hat{f} = f(x^*)$ is the corrupted objective evaluated at the true signal $x^*$;

**3) Stepsize.** We set the parameter $\gamma = \alpha_0^{-1} \sqrt{K/m}$ where $m$ is the batch size; For synthetic dataset, we test 10 evenly spaced $\alpha_0$ values in range $[10^{-1}, 10^2]$ on logarithmic scale, and for `zipcode` dataset we set such range of $\alpha_0$ to $[10^1, 10^3]$;

**4) Maximum iteration.** We set the maximum number of epochs to be 200 and 400 respectively for minibatch and momentum related tests;

**5) Batch size.** We take minibatch size $m$ from the range $\{1, 4, 8, 16, 32, 64\}$;

**6) Sub-problems** The solution to the proximal sub-problems is left in the appendix.

For each algorithm, speedup from minibatching is quantified as $T_1^*/T_m^*$ where $T_m^*$ is the total number of iterations for reaching the desired accuracy, with batch size $m$ and the best initial stepsize $\alpha_0$ among values specified above. Specially, if an algorithm fails to reach desired accuracy after running out of 400 epochs, we set its iteration number to the maximum.

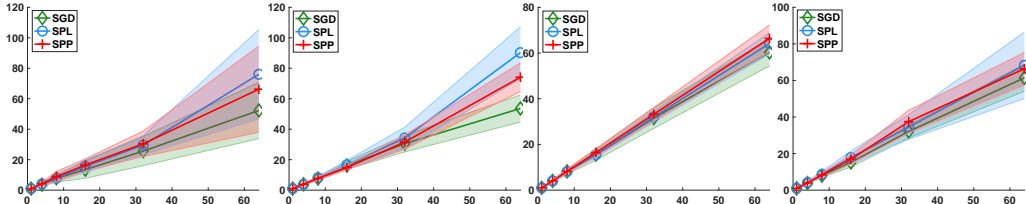

Figure 1: Speedup over minibatch sizes. The left two are for synthetic datasets $\kappa = 10, p_{\text{fail}} \in \{0.2, 0.3\}$; Digit datasets: digit image (id:24) with $p_{\text{fail}} \in \{0.2, 0.3\}$.

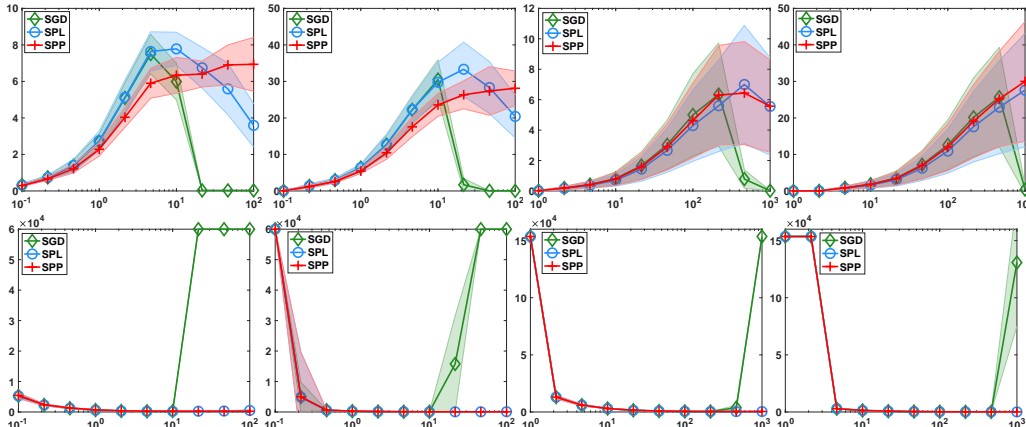

Figure 2: From left to right: synthetic datasets with $m \in \{8, 32\}$ and `zipcode` image (id=24) with $m \in \{8, 32\}$. x-axis: initial stepsize $\alpha_0$. y-axis (first row): speedup over the sequential version: $T_1^*/T_m^*(\alpha_0)$ where $T_m^*(\alpha_0)$ stands for the number of iterations when using batch size $m$ and initial stepsize $\alpha_0$. y-axis (second row): Total number of iterations.

Figure 1 plots the speedup of each algorithm over different values of batch size according to the average of 20 independent runs. It can be seen that SPL exhibits a linear acceleration over the batch size, which confirms our theoretical analysis. Moreover, we find SGD admits considerable acceleration using minibatches, and sometimes the speedup performance matches that of SPL and SPP. This observation seems to suggest the effectiveness of minibatch SGD in practice, despite the lack of theoretical support.

Next, we investigate the sensitivity of minibatch acceleration to the choice of initial stepsizes. We plot the algorithm speedup over the initial stepsize $\alpha_0$ in Figure 2 (1st row). It can be readily seen that SGD, SPL and SPP all achieve considerable minibatch acceleration when choosing the initial stepsize properly. However, SPL and SPP enjoy a much wider range of initial stepsizes for good speedup performance, and hence, lays more robust performance than SGD. To further illustrate the robustness of SPL and SPP, we compare the efficiency of both algorithms in the minibatch setting. In contrast to the previous comparison on the relative scale, we directly compare the iteration complexity of the two algorithms. We plot the total iteration number over the choice of initial stepsizes in Figure 2 (2nd row) for batch size $m = 8$ and 32. We observe that minibatch SPL(SPP)s exhibits promising performance for a wide range of stepsize policies, while minibatch SGD quickly diverges for large stepsizes. Overall, our experiment complements the recent work [9], which shows that SPL (SPP) is more robust than SGD in the sequential setting.

Our second experiment investigates the performance of the proposed momentum methods. We compare three model-based methods (SGD, SPL, SPP) and extrapolated model-based methods (SEGD, SEPL, SEPP). We generate four testing cases: the synthetic datasets with $(\kappa, p_{\text{fail}}) = (10, 0.2)$ and

$(10, 0.3)$; `zipcode` with digit images of id 2 and $p_{\text{fail}} \in \{0.2, 0.3\}$. We set $\alpha_0 \in [10^{-2}, 10^0], \beta = 0.6$ for synthetic data, and set $\alpha_0 \in [10^0, 10^1], \beta = 0.9$ for `zipcode` dataset. The rest of settings are the same as in minibatch with $m = 1$.

Figure 3 plots the number of epochs to $\varepsilon$-accuracy over initial stepsize $a_0$. It can be seen that with properly selected momentum parameters (SEGD, SEPL, SEPP) all suggest improved convergence when stepsize is relatively small.

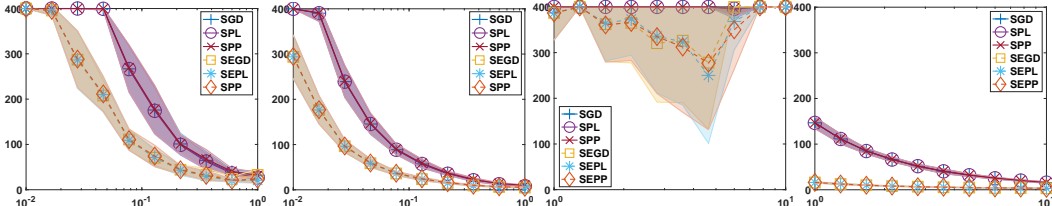

Figure 3: From left to right: synthetic datasets with $\kappa = 10, p_{\text{fail}} \in \{0.2, 0.3\}, \beta = 0.6$ and `zipcode` image (id=2) with $p_{\text{fail}} \in \{0.2, 0.3\}, \beta = 0.9$. x-axis: initial stepsize $\alpha_0$. y-axis: number of epochs on reaching desired accuracy

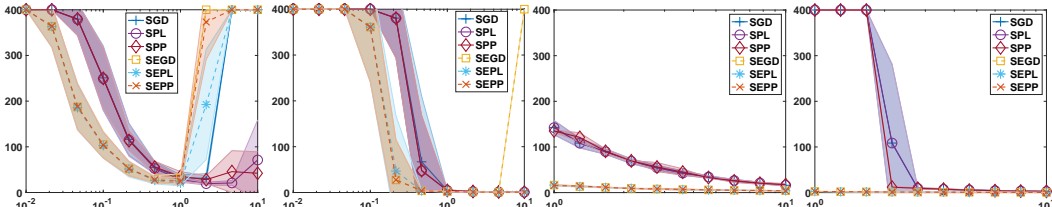

Figure 4: From left to right: synthetic datasets with $\kappa = 10, p_{\text{fail}} = 0.2, \beta = 0.6, m \in \{1, 32\}$ and `zipcode` image (id=24) with $p_{\text{fail}} = 0.3, \beta = 0.9, m \in \{1, 32\}$. x-axis: initial stepsize $\alpha_0$. y-axis: number of epochs for reaching desired accuracy

In the last experiment, we attempt to exploit the performance of the compared algorithms when minibatching and momentum are applied simultaneously. The parameter setting is the same as that of the second experiment, except that we choose $m \in \{1, 32\}$. Results are plotted in Figure 4 and it can be seen that minibatch `SMOD`, when combined with momentum, exhibits even better convergence performance and robustness.

## 7 Discussion

On a broad class of non-smooth non-convex (particularly, weakly convex) problems, we make stochastic model-based methods more efficient by leveraging minibatching and momentum—two techniques that are well-known only for `SGD`. Applying algorithm stability for optimization analysis is a key step to achieving improved convergence rate over the batch size. This perspective appears to be interesting for stochastic optimization in a much broader context. Although some progress is made, we are unable to show whether minibatches can accelerate `SGD` when the objective does not have a smooth component. Note that the complexity of `SGD` already matches the best bound of full subgradient method. It would be interesting to know whether this bound for `SGD` is tight or improvable. It would also be interesting to study the lower bound of `SGD` (and other stochastic algorithms) in the non-smooth setting. Some interesting recent results can be referred from [22, 33].

## 8 Acknowledgement and disclosure of funding

The authors are grateful to the Area Chairs and the anonymous reviewers for their constructive suggestions. QD was partially supported by National Natural Science Foundation of China (Grant 11831002, 72150001).

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
