# OpenReview forum: "Minibatch and Momentum Model-based Methods for Stochastic Weakly Convex Optimization"
_NeurIPS.cc/2021/Conference — NeurIPS 2021 Poster_

### Official Review · Reviewer_cuoE · 2021-06-30

**Rating:** 3
**Confidence:** 4

**Summary:**

This contribution introduces and analyzes mini batch variants of stochastic model-based optimization methods when the loss function is possibly nonconvex and non smooth. Their algorithms are accompanied with a theoretical analysis and numerical experiments.

**Ethical Concerns:**

N.A.

**Ethics Review Area:**

["I don’t know"]

**Limitations And Societal Impact:**

No they have not.

**Main Review:**

Overall, I don’t understand what makes those methods different from classical stochastic optimization method. All of those baselines are model based in the sense that the updated parameters throughout the iterations, depend on the loss function and the model employed.
Hence, Algorithm 1 boils down to any proximal and regularized gradient step, which has been around for a long time.
It would be greatly appreciated if the authors can clarify what makes their method different from what is in the literature.

- Numerical experiments: The authors only compare, in Figure 1 and 2, their method with plain SGD.
Given the sheer size of optimization methods, one would expect much more baselines for fair comparison.

Minor typos:
- In the checklist: « Did you include the license to the code and datasets? [Yes] See Section ??. » what is the section number?
- Using the same notation for the stochastic objective function and the unregularized one is confusing.
- Algorithm 1 writing requires more details: How is the stepsize initialized, how are the m_k decided, what is the output of the method.


**Time Spent Reviewing:**

1

---

> ### Author Response · Authors · 2021-08-10
> **Response to Reviewer cuoE**
>
> ### Reviewer cuoE
>
>  **Question 1**
>
> > I don’t understand what makes those methods different from classical stochastic optimization method.... Algorithm 1 boils down to any proximal and regularized gradient step, which has been around for a long time. It would be greatly appreciated if the authors can clarify what makes their method different from what is in the literature.
>
> The main difference is that the classic SMOD is sequential in nature while the new algorithms can process a minibatch of data in parallel. The key point is that we  design a new stepsize/learning rate adapted to the minibatch SMOD, and we  justified that the new approach does achieve an improved convergence rate. Furthermore, we empirically confirm the effectiveness of our approach in the experiments.
>
> > Numerical experiments: The authors only compare, in Figure 1 and 2, their method with plain SGD. Given the sheer size of optimization methods, one would expect much more baselines for fair comparison.
>
> Figure 1 and 2 aim at testing the effectiveness of minibaching for the SMOD family. (We did test momentum and minibatch SGD in Figure 4, though). While there are many more advanced SGD variants (e.g. adaptive gradient), it is indeed out of the scope of this paper and adding more baselines will increase the noise in measuring the performance.
>
> Moreover, our experiment setup is similar to those from the work in the same line ([1,2,3]) on SMOD; these works have been published in decent venues like NeurIPS, ICML and SIAM OPT. Hence, we are confident that our choice of the compared algorithms suffices to confirm our theoretical analysis.
>
> We sincerely request the reviewer to re-evaluate our score if he/she feels that the concern is addressed.
>
> **Minor Typos**:
>
> > In the checklist: « Did you include the license to the code and datasets? [Yes] See Section ??. » what is the section number?
>
> This is the instruction block, not our response.  We are sorry for the confusion and will remove it in an updated version.
>
> > Using the same notation for the stochastic objective function and the unregularized one is confusing.
>
> It is unclear to us what "unregularized  one" refers to.  However, since our notation does not bother the other reviewers, we will keep the notations for consistency,  but would gladly explain more to reviewer cuoE if necessary.
>
> > Algorithm 1 writing requires more details: How is the stepsize initialized, how are the m_k decided, what is the output of the method.
>
> Stepsize and  batch size $\\{m_k\\}$​​​​​​ are hyper-parameters and hence are unspecified in the pseudo-code.  As for the parameter setting,  they are available in Theorem 3.4. We understand the reviewer's concern since a default parameter will be given in many practical paper when the algorithm is highly tuned for a very specific dataset or model such as deep learning,  e.g. Algorithm 1 of [4]. However,  it is unusual for a theoretical paper to specify these parameters  in the pseudo-code as the problem is abstract.  For example, see Algorithm 3.1 of [1] for the style. Due to the above concerns, we insist on the keeping these two parameters unspecified.
>
>
>
> **References**
>
> [1] Davis and Drusvyatskiy. Stochastic model-based minimization of weakly convex functions. SIOPT, 2019.
>
> [2] V. Mai and M. Johansson. Convergence of a stochastic gradient method with momentum for non-smooth non-convex optimization. In Proceedings of the 37th International Conference on Machine Learning, pages 6630–6639, 2020.
>
> [3] H. Asi, K. Chadha, G. Cheng, and J. C. Duchi. Minibatch stochastic approximate proximal point methods. Advances in Neural Information Processing Systems, 33, 2020.
>
> [4] Kingma, D. P., & Ba, J. (2014). Adam: A method for stochastic optimization. *arXiv preprint arXiv:1412.6980*.

---

> > ### Comment · Reviewer_cuoE · 2021-08-16
> > **Thank you for the rebuttal**
> >
> > Thank you for the replies.
> > While those precisions help the understanding, I am not changing the score given important flaws in the paper (at the current stage)

---

> > > ### Author Response · Authors · 2021-08-27
> > > **Reply**
> > >
> > > We thank the reviewer for the response. But we are still confused by the reviewer's comment on the "important flaws." We have the feeling that the contributions and the technical novelty of our work are clearly addressed.  If the reviewer has more specific requests, we would be very glad to discuss.

---

### Official Review · Reviewer_Kosw · 2021-07-15

**Rating:** 6
**Confidence:** 5

**Summary:**

The paper presents several minibatch and momentum extensions of stochastic model-based optimization methods (SMOD) for learning with non-smooth and weakly convex loss functions. For SMOD with minibatch size $m$ and iteration count $K$, the authors showed via a uniform stability argument that the algorithm converges to stationary point at the rate of $O(\frac{1}{\sqrt{mK}})$. Similar convergence rates have been established for an extrapolated variant of SMOD with or without minibatching. The actual performance of the proposed methods is evaluated in a set of experiments on synthetic and real-data robust phase retrieval.

**Limitations And Societal Impact:**

Yes,  the authors have partially addressed the limitations and potential negative societal impact of their work.

**Main Review:**

The paper is technically sound and the main results make sense to me. I am particularly impressed with the clear and clean writing style, and how all steps in the proofs are carefully explained. However, I still have some reservations on the novelty and significance of results as specified below:

- Originality: While interesting, the idea of extending SMOD-type methods to minibatch optimization using stability argument is definitely not new. For example, for distributed learning with convex and Lipschitz losses, the benefit of minibatching has already been revealed for stochastic proximal point (SPP) method through the lens of uniform algorithmic stability theory [1]. More precisely, up to minor differences, Theorem 3.2 is identical the Lemma 2 in [1]. Therefore, the only novel point I can spot in this part is using the generalization bounds in Theorem 3.2 to analyze the convergence of SMOD for weakly convex losses, as summarized in Theorem 3.3 & 3.4. The technical part of these theorems is more or less straightforward given the techniques developed in [2] (i.e., reference [8] in the paper). Overall, the degree of novelty of the minibatch extension is fairly low with respect to the prior work in this line. The proposed momentum extension is relatively more impressive, and somewhat novel as far as I know about.

- Significance of results: My main concern regarding the proposed extrapolated variant of SMOD is the strength of results in Theorem 4.2 & 4.3, which seem showing no significant improvement over the vanilla version. Since the method reduces to a heavy-ball acceleration strategy in the SGD case, it is not surprising that it does not substantially improve the rate of convergence for generic convex or non-convex losses. On the other side, it is well known that heavy-ball is beneficial for accelerating batch GD for quadratic programs. Then a natural question is if it is possible for the extrapolated SMOD method to achieve acceleration at least for quadratic losses? Anyway, a careful discussion on the advantages of the momentum extension is highly desirable to justify the usage of method in theory.

Minor comments:

- Clarity: Since Assumption A6 & A7 are of very limited interest unless $M$ and $D$ are extremely small, it is suggested to carrying out the analysis without introducing these void assumptions so that the core results can be presented in a neater way.

- Experiment: As the comparison is made among stochastic optimization methods, it is desirable to provide error bars in the plots to illustrate the variance of computation.

[1] Wang, Wang, and Srebro. Memory and communication efficient distributed stochastic optimization with minibatch prox. COLT, 2017.

[2] Davis and Drusvyatskiy. Stochastic model-based minimization of weakly convex functions.  SIOPT, 2019.



=== Post rebuttal update ===

The major concerns raised in my initial review on the momentum extension have been much clarified in the author response. It's good to know, based on the augmented results provided during author discussion phase, that the robustness of SMOD to the choices of step-size can be much preserved by its momentum variant. As for the minibatching acceleration analysis, while  the results for weakly-convex problems are somwhat interesting, IMO the technical contribution of this part is too incremental for NeurIPS given the prior work of Wang et al. (2017) and Davis & Drusvyatskiy (2019).  It is suggested to tune down the claim of this part accordingly.

 Overall, I would still like to change my rating one grid up to 6, apprecaiting the novelty and strength of the momentum extension/analysis of SMOD.


**Time Spent Reviewing:**

11

---

> ### Author Response · Authors · 2021-08-10
> **Response to Reviewer Kosw**
>
> ### Reviewer Kosw
>
> We thank the reviewer and appreciate the time spent on our paper.
>
> **Originality on Minibatch**
>
> We thank the reviewer for pointing out the interesting work [1] that uses a similar technique in distributed optimization. Indeed, the work [1] applied stability analysis and their key lemma is very similar to our Theorem 3.2.  We are a bit surprised, though, that the work [1] was not cited in our main referred work on minibatching. We will gladly refer to this work and certainly acknowledge its contribution.
>
> Technically, we agree with the reviewer that the analysis on the minibatch part is not surprising once the connection between stability and optimization is made. However, it is certainly non-trivial for us since we were not aware of [1] and came to the path from completely different motivation. A theoretical motivation is that we hope to fill in the complexity gap between stochastic SMOD and deterministic SMOD in the literature [2, 8], and we find that stability is a more effective tool than the standard  Lipschitz argument for nonsmooth optimization.  We acknowledge the contribution of [1], however, we feel that the use of stability analysis is not well-known by the optimization field, or at least by our referred theoretical work. It certainly should deserve more attention. We sincerely believe that our paper will pass this important message to the NeurIPS/Optimization community and provoke interesting thoughts.
>
> In addition, there are several important distinctions in the studied problems and algorithms.
>
> 1. The work [1] proposed a distributed stochastic proximal point method where the subproblem is approximately solved by a distributed stochastic gradient method such as SVRG. Their main goal is to use minibatch acceleration to find a desirable balance between communication and memory resources. In contrast, we characterize a class of algorithms/problems that are "minibatching-friendly" with a unified convergence analysis;  our analysis also provides some insights on when linear speedup is achievable via minibatching and the potential limitations in stochastic subgradient.
>
> 2. The work [1] develops complexity rates for convex and  strongly convex optimization while our paper  develops complexity  for both non-convex  (main article) and convex optimization (please also check Appendix C). We also hightlight that in the appendix we extend our analysis to SMOD with Nesterov momentum and obtain the optimal rate of convergence.
>
> 3. Besides the technical part, we want to highlight the significance of our results for stochastic optimization. The convergence of sequential SMOD, and the advantage of SMOD over SGD has been established by [2], yet it was still unclear how to make SMOD practical in the modern parallel architecture. Our work gives a much clear picture on when SMOD can be accelerated by minibatching and where the limitation is. We believe that the convergence result itself will be highly interesting to the theoretical community in NeurIPS. From a practical side, our paper will certainly benefit those practitioners who are willing to seeking robust alternatives of SGD in real applications.
>
>
> **Significance of Momentum**
>
> We thank the reviewer for acknowledging the novelty of our momentum methods. We agree that our extrapolated method indeed does not have better rate. However, we want to address that the main challenge here is not to show an improved convergence rate but actually to show convergence rate comparable to the standard SGD. While this rate is no surprise, it is indeed nontrivial.  Without careful analysis, one would easily see the challenge that we can show convergence only if the momentum term is set zero (i.e. $\\beta=0$​​).  Hence we believe that our work on extrapolated SMOD significantly broadens the set of algorithms where the momentum technique works and is indeed worth publishing in NeurIPS. Indeed, to show convergence or even to show further acceleration, most existing work relies on smoothness assumption  (e.g. [4,5]) or has to be restricted to some specific algorithm (such as projected subgradient [6]). And these works are all published in the celebrated conferences like NeurIPS or ICML.
>
> Showing that heavy ball accelerated convergence on quadratic problems might be challenging. This requires us to substantially improve our analysis for convex problem in Appendix C.  However, since our paper already has rich contents, we feel that it is more appropriate to leave the extension as future work.
>
> To summarize,  we believe that our technical novelty and significance indeed meet the standard of NeurIPS publication and we sincerely wish that the reviewer increase the score if our response addresses those concerns.
>
>
>
> **Minor Comments**
>
> 1. Indeed, we feel that the additional assumptions A6 and A7 are non-standard and may cause confusions. In a later revision, we will remove A6 and A7 and just state the result under A5. Consequently, in Lemma 3.1, the  $\\varepsilon$​ stability will be defined with $\\varepsilon=\\frac{2L^2}{m(\\gamma-\\lambda)}$​. In Theorem 3.4, the bound(10) will have no $C_2$​ term, and parameter $\\eta$​ will be set to $\\eta=L(\\frac{2\\rho K}{m\\Delta})^{1/2}$​. This gives us a rate of the form $O(1/K+1/\\sqrt{mK})$​. Thus our main result still holds.
>
> 2. We did not plot the error bar in our initial version since we mostly measured convergence in epoch and the deviation seems small in most cases. In our revised version we promise to add error bar and below is a sample table of the quantile of iteration/epoch number for SPL with $p_{\\text{fail}} = 0.3, \\beta = 0.9, \\text{id=24}, m = 32$​​​​​ over 30 repetitions.
>
>    |$\\alpha_0$ | $10^{2 / 9}$ | $10^{1 / 3}$ | $10^{4/9}$ | $10^{5/9}$ | $10^{6/9}$ | $10^{7/9} $| $10^{8/9}$ | $10^1$|
>    |:--:|:--:|:--:|:--:|:--:|:--:|:--:|:--:|:--:|
>    |Q25%-SPLIter | 3.1e+05 | 8.5e+03 | 6.7e+03 | 5.0e+03 | 4.1e+03 | 3.1e+03 | 2.4e+03 | 1.8e+03 |
>    |Mean-SPLIter | 3.1e+05 | 8.3e+04 | 7.1e+03 | 5.5e+03 | 4.4e+03 | 3.3e+03 | 2.6e+03 | 1.9e+03 |
>    |Q75-SPLIter | 3.1e+05 | 1.5e+05 | 7.5e+03 | 5.9e+03 | 4.7e+03 | 3.5e+03 | 2.7e+03 | 2.1e+03 |
>    |Q25%-SPLEpc | 400 | 11 | 9 | 7 | 5 | 4 | 3 | 2 |
>    |Mean-SPLEpc | 400 | 108 | 9 | 7 | 6 | 4 | 3 | 2 |
>    |Q75%-SPLEpc | 400 | 206 | 19 | 8 | 6 | 5 | 4 | 3 |
>
>    **Explanation of the table**
>
>    Q25%-SPLIter/Epoch: the lower 25% quantile of the SPL iteration/epoch counts
>
>    Mean-SPLIter/Epoch: mean of the SPL iteration/epoch counts
>
>    Q75%-SPLIter/Epoch: the upper 75% quantile of the SPL iteration/epoch counts
>
>
>
> **References**
>
> [1] Wang, Wang, and Srebro. Memory and communication efficient distributed stochastic optimization with minibatch prox. COLT, 2017.
>
> [2] Davis and Drusvyatskiy. Stochastic model-based minimization of weakly convex functions. SIOPT, 2019.
>
> [3] S. Ghadimi, G. Lan, and H. Zhang. Mini-batch stochastic approximation methods for nonconvex stochastic composite optimization. Mathematical Programming, 155(1-2):267–305, 2016.
>
> [4] I. Gitman, H. Lang, P. Zhang, and L. Xiao. Understanding the role of momentum in stochastic gradient methods. In Advances in Neural Information Processing Systems,
>
> [5] Liu, Yanli, Yuan Gao, and Wotao Yin. 2020. “An Improved Analysis of Stochastic Gradient Descent with Momentum.” In Advances in Neural Information Processing Systems, 33:18261–71.
>
> [6] V. Mai and M. Johansson. Convergence of a stochastic gradient method with momentum for non-smooth non-convex optimization. In Proceedings of the 37th International Conference on Machine Learning, pages 6630–6639, 2020.
>
> [7] H. Asi, K. Chadha, G. Cheng, and J. C. Duchi. Minibatch stochastic approximate proximal point methods. Advances in Neural Information Processing Systems, 33, 2020.
>
> [8] D. Drusvyatskiy and C. Paquette. Efﬁciency of minimizing compositions of convex functions and smooth maps. Mathematical Programming, pages 1–56, 2018.

---

> > ### Comment · Reviewer_Kosw · 2021-08-18
> > **Response to Authors**
> >
> > Thank you for the detailed responses. While the minibatching results are interesting in the context of stochastic weakly convex optimization, I still think the strength of technical contribution in this part is somewhat below the threshold given the prior work of Wang et al. (2017) and Davis & Drusvyatskiy (2019). As for the momentum part, since the rate of convergence shows no clear advantage over that of the extrapolated SGD, it might be considered to additionally carry out a robustness analysis to more convincingly justify the reason of using extrapolated SMOD.

---

> > > ### Author Response · Authors · 2021-08-20
> > > **A robust analysis for SMOD**
> > >
> > > We thank the reviewer for the quick response and the constructive comments, which we believe helps us further improve the paper.
> > >
> > > The reviewer suggests that *we carry out a robustness analysis to more convincingly justify the reason for using extrapolated SMOD*. We remark that the reason of using extrapolated SMOD is to infuse momentum for more aggressive exploration of the parameter space, rather than for enhancing robustness. Our experiments indeed verify the effectiveness of momentum.
> > >
> > > However, in this reply, we want to draw the attention of the reviewer to the fact that **extrapolated SMOD indeed is more robust than SGD**. This is another contribution of our paper, which will be added in our revised version accordingly.
> > >
> > > Putting it in the context, we argue that in the *convex setting* and under mild conditions (Assumption A8, Line 597 in the appendix), extrapolated SMOD can guarantee the boundedness of the iterates. In contrast, without strong assumptions such as Lipschitz continuity or bounded domain, this property is not possessed by subgradient method in the non-smooth setting.  We kindly remind the reviewer that A8 is analogous to A5: A8 is naturally satisfied by SPL and SPP in the convex setting; it is also satisfied by SGD when the objective is smooth (composite).
> > >
> > > Following the robustness analysis in [AD2019]  (where it is dubbed stability analysis), we want to show that $\\mathbb{E}[\\text{dist}(x^k,\\mathcal{X}^*)]<+\\infty$  where $\\mathcal{X}^*$ is the set of optimal solutions. This guarantees that with probability one, the iterates are bounded.
> > >
> > > For simplicity, we highlight the key difference between standard SGD and extrapolated SMOD (under A8) .
> > >
> > > **1.** SGD (which takes $x^{k+1}=\\text{argmin}_x \\langle f^\\prime(x^k,\\xi_k),x \\rangle +\\frac{\\gamma_k}{2}\\Vert x-x^k\\Vert^2$) .
> > >
> > >    From standard analysis (e.g. [Nemirovski et al] ), it is very easy to show
> > >
> > >    $$\\mathbb{E}\\,[\\Vert x^{\\ast} - x^{K + 1}\\Vert^2]\\le \\Vert {x}^\\ast - x^0\\Vert^2+\\sum_{k=0}^K\\frac{1}{\\gamma_k^2}\\mathbb{E}[\\Vert f^\\prime(x^k,\\xi_k)\\Vert^2]$$
> > >
> > > **2.** But in extrapolated SMOD, we are able to show
> > >    $$\\mathbb{E} [\\| x^{\\ast} - x^{K + 1} \\|^2] \\leq  \\| {x}^\\ast - x^0
> > >       \\|^2 + \\beta (1 - \\beta) \\| x^1 - x^0 \\|^2 +
> > >       \\sum_{k = 0}^K \\frac{2}{\\theta^2 \\gamma_k^2}\\mathbb{E} [\\| f' (x^{\\ast}, \\xi_k) \\|^2],$$
> > >
> > >
> > > and recall that we set $\\theta=1-\\beta$.  For the sake of asymptotic analysis,  stepsize parameter  $\\gamma_k$ in extra-SMOD is now indexed by $k$.
> > >
> > > We add an anonymous link below to the proof of  the above boundedness result.
> > >
> > > > https://anonymous.4open.science/r/neurips_11182_proof-38CF
> > >
> > > Some observations are in order.
> > >
> > > 1. To show that SGD has bounded iterates, one often needs to assume Lipschitzness or bounded domain, which ensures that $\\Vert f^\\prime(x^k,\\xi_k)\\Vert$ is bounded above. However, when  $\\Vert f^\\prime(x^k,\\xi_k)\\Vert$ is unbounded or the bound is large (e.g. the gradient norm of exponential function), we need sufficiently large $\\{\\gamma_k\\}$ (i.e. small stepsize $1/\\gamma_k$) to ensure the boundedness of $x^{k+1}$. In contrast,  for SMOD the bound only depends on the subgradient over the optimal solutions $x^*$. In many problems, (e.g. interpolation setting),  $\\| f' (x^{\\ast}, \\xi) \\|$ will be substantially smaller than $\\| f' (x, \\xi) \\|$ over the feasible domain. Therefore, extrapolated SMOD is more robust because it has boundedness guarantee over larger stepsize ($1/\\gamma_k$) options
> > > 2. We also note that the best bound for SMOD is when $\\beta=0, \\theta=1$,  which implies that adding momentum may increase the instability. This is indeed expectable, because adding momentum gives the algorithm more chance of exploration through the parameter space, but at the cost of potentially moving away from the original solution path.
> > >
> > > Our new result further improves the contribution of our paper and appears to address the concern of the reviewer. We sincerely hope that the review can increase the score based on our new result.
> > >
> > >
> > > **References**
> > >
> > > [Nemirovski et al] Nemirovski, A., et al. “Robust Stochastic Approximation Approach to Stochastic Programming.” Siam Journal on Optimization, vol. 19, no. 4, 2008, pp. 1574–1609.
> > >
> > > [AD2019] Asi, Hilal, and John C. Duchi. "Stochastic (approximate) proximal point methods: Convergence, optimality, and adaptivity." *SIAM Journal on Optimization* 29.3 (2019): 2257-2290.

---

> > > > ### Comment · Reviewer_Kosw · 2021-08-26
> > > > **Response to the additonal robustness analysis**
> > > >
> > > > Thank you for your further providng additional results on the iteration-stability of momentum SMOD. It's good to see that the robustness of SMOD to the choices of step-size can be much preserved by its momentum variant, though the best bound occurs at $\beta=0$.  The stability analysis seems easily generalizable to the minibatching case. I thus would upgrade my score to a positive 6 to appreciate the momentum section of this work. .

---

> > > > > ### Author Response · Authors · 2021-09-03
> > > > > **Thank you for your insightful comments**
> > > > >
> > > > > We sincerely thank the reviewer for updating the score as well as the constructive comments which truly helped improve our paper !!

---

### Official Review · Reviewer_37oF · 2021-07-16

**Rating:** 4
**Confidence:** 5

**Summary:**

The paper proposed stochastic model-based method and its momentum version for solving non-smooth non-convex optimization problems. The convergence results of the proposed methods are established. The authors also provide some experimental results for verifying the effectiveness of the proposed methods.

**Limitations And Societal Impact:**

No. The new assumption A5 is never used in the lierature.

**Main Review:**

Originality: The convergence analysis seems non-trivial.

Quality: The assumptions are too strong. For example, it seems that Assumption 5 is never used in the literature. How to justify it?

Clarity: The paper is well organized, and the writing is clear to me.

Significance: The “improved” convergence result of SMOD requires additional assumptions (e.g., A5, A6, A7), which makes the contribution of this paper limited. With Assumption 5, having such convergence result is not surprising. The Assumption 5 makes the results no general enough and is not applicable to standard stochastic subgradient methods for solving non-smooth problems.

**Time Spent Reviewing:**

3

---

> ### Author Response · Authors · 2021-08-10
> **Response to Reviewer 37oF**
>
> ###  Reviewer 37oF
>
> **Significance**
>
> > Significance: The “improved” convergence result of SMOD requires additional assumptions (e.g., A5, A6, A7), which makes the contribution of this paper limited. With Assumption 5, having such convergence result is not surprising. The Assumption 5 makes the results no general enough and is not applicable to standard stochastic subgradient methods for solving non-smooth problems.
>
> A6 and A7 can be removed without affecting the main conclusion. See Remark 3, 4 in our apper. Also see our reply to reviewer Kosw.
>
>
>
> ### **[Update 8/17]**
> Reviewer 37oF argued that A5 has some limitations: 1) A5 is not general enough and 2)  A5 is not applicable to stochastic subgradient.
>
> We can not agree with reviewer 37oF on the first one. Please read Remark 2 of our paper where we have already argued about the generality of A5. Second, while we agree that A5 is not applied to stochastic subgradient, we do not think it will impair the significance of our contribution. We argue from two perspectives.
>
> 1. On the practical side, whether A5 applies to stochastic subgradient is not our major concern. As we already mentioned, we study SMOD due to its robustness. From this point of view, SPL and SPP are better (robust) choices in the SMOD family since they are robust to the hyperparameter tuning. Please also read [8]. Hence, to  practically promote SMOD in terms of robustness, it is more important to speed up these two algorithms rather than SGD by minibatching. Since A5 guarantees that both SPL and SPP benefit from minibatching, (and of course any other algorithms satisfying A5 would),  we believe that our contribution is justified, and the limitation mentioned by the reviewer is not a concern of our paper.
>
> 2. On the theoretical side, the limitation mentioned by the reviewer should be expectable. We remind the reviewer 37oF that, at least to the best of our knowledge,  the best rate of stochastic subgradient method matches the rate of deterministic subgradient method, which means theoretically, using minibatch does not improve convergence.  In fact, this is already known in the convex case, since the worst case complexity of both methods are $O(1/\sqrt{K})$ [B2015].   Therefore, we argue that, without additional assumptions, it is unlikely to accelerate the stochastic subgradient method by minibatching.
>
>
>
> We also disagree with reviewer 37oF claiming that the new assumption A5 is never used in the literature. As we already stated in the paper, applications are enormous. For example, Prox-SGD for smooth composite problems (e.g. Lasso) fall in this category.  Other examples include blind deconvolution, phase retrieval, matrix completion and many many others. (e.g. [8]). We strongly and sincerely request the reviewer 37oF to refer to the work [7, 8]. For example, see the use of an assumption similar to A5 in page 4 [7].
>
> **References**
>
> [7] V. Charisopoulos, Y. Chen, D. Davis, M. Díaz, L. Ding, and D. Drusvyatskiy. Low-rank matrix recovery with composite optimization: good conditioning and rapid convergence. Foundations of Computational Mathematics
>
> [8] D. Davis and D. Drusvyatskiy. Stochastic model-based minimization of weakly convex functions. Siam Journal on Optimization, 29(1):207–239, 2019.
>
> [B2015] Bubeck, Sébastien. "Convex optimization: Algorithms and complexity." *arXiv preprint arXiv:1405.4980* (2014).

---

### Official Review · Reviewer_kmZz · 2021-07-16

**Rating:** 6
**Confidence:** 4

**Summary:**

The paper studied stochastic model based methods (SMOD), and prove complexity bounds for minibatch SMOD and momentum SMOD.


**Limitations And Societal Impact:**

See above for limitations and no negative societal impact.

**Main Review:**

Strength: the paper proves complexity bounds for SMOD with minibatch and momentum, and show that minibatch provides speedup, which is new to the literature.


The complexity result, such as Theorem 3.4, indeed implies  linear speedup, however only for m being not too large. Suppose $ m \geq 1/\epsilon^2 $, then the complexity would be $ T_\epsilon = O(\Delta/\epsilon^2) $ which is independent of batch size. As a result, if would be interesting to see in Figure 1, what the plots will be like for even large $m$. In addition, what is the “desired accuracy’’ chosen in the numerical experiments, will the value of accuracy affects the linear speedup of minibatch?

Minor:
Line 12, “…possibly improve…’’ please remove or rewrite this statements since it’s meaningless. Even in the deterministic setting, momentum is not guaranteed to provides acceleration. However, it might leads to smaller objective function values. It would be good if the authors can check if this is true for the considered problems here.

Line 149, “B = ”, equal sign missing.

Line 205, “Despite such a relation’’, a missing


**Time Spent Reviewing:**

4

---

> ### Author Response · Authors · 2021-08-10
> **Response to Reviewer kmZz**
>
> ###  Reviewer kmZz
>
> We thank the reviewer for the efforts reviewing the paper.
>
> **Experiments**
>
> >  ... if would be interesting to see in Figure 1, what the plots will be like for even large m.
>
> Indeed, the increase of speedup over $m$​​ tends to slow down for large batchsize $m$​​​​. To see the intuition, we extend our study in Figure 1 by plotting the  the minibatch speedup for SGD/SPL with $\\kappa=10, p_\\text{fail}=0.3$​ and larger batchsizes $m\\in\\{1, 64, 128, 240, 260, 280\\}$​​​​ over 10 repetitions. It can be seen that as $m$​​​​ grows, the complexity is less dependent on the batchsize.
>
> |      m       |  1   |  64  | 128  | 240  | 260  | 280  |
> | :----------: | :--: | :--: | :--: | :--: | :--: | :--: |
> | SGD. Speedup |  1   |  59  |  91  | 141  | 136  | 131  |
> | SPL. Speedup |  1   |  90  | 200  | 300  | 300  | 300  |
>
>
>
> > what is the “desired accuracy’’ chosen in the numerical experiments,
>
> We terminate the algorithm when the objective value is below a target level. Since the corrupted problem is non-convex and we have no access to the optimal value,  we set $f(x^k)\\le 1.5 f (\\hat{x})$​  as the termination criterion, here the reference point $\\hat{x}$​​  is the optimal solution of the problem with no corruption. Practically $f (\\hat{x})$​​​​ is a small quantity and we find that the solution satisfying our criterion also gives a high-quality solution (please refer to the additional experiments for the Blind Deconvolution and also the examples of image recovery in the appendix).
>
> > Will the value of accuracy affect the linear speedup of minibatch?
>
> For our experiment setting, the answer is yes. In general, SMOD has a sublinear rate of convergence and exhibits great performance for obtaining a medium-level accuracy. However, we find that if the target accuracy is overly high, SMOD will not terminate in the maximum number of iterations no matter how the batchsize/stepsize is tuned.
>
> **Minor Issues**
>
> > “…possibly improve…’’ please remove or rewrite this statements since it’s meaningless.
>
> We agree with the reviewer that we did not show faster convergence of momentum method. In the revision we will write "we propose a new stochastic extrapolated model-based method and establish its rate of convergence in the non-smooth and non-convex setting".
>
> > Momentum ... it might leads to smaller objective function values. It would be good if the authors can check if this is true for the considered problems here.
>
> The answer is yes, since in the experiments the objective function value is exactly the stopping criterion we set.
>
> **Typos**
>
> We thank the reviewer for pointing out the typos and we will fix them (in Line 149 and 205) as suggested by the reviewer.

---

> > ### Comment · Reviewer_kmZz · 2021-08-24
> > **Response to Authors**
> >
> > Thanks for the clarification, looks good to me.

---

### Decision · Program_Chairs · 2021-09-28

**Decision:**

Accept (Poster)

**Comment:**


The paper proposes an analyses of model based methods with mini-batching and momentum. The topic is relevant to the Neurips community. But the reviewers were unclear as to how this fits in and compares to existing work, in particular due to Assumption 5. On that note, since most of the models involve a linearization (excluding the proximal point method) Assumption 5 is essentially smoothness. This makes it a bit confusing given the emphasis of the work on the non-smooth setting. Reviewer Kosw also raised questions as to the technical novelty of the mini-batch analysis based on stability analysis, given closely related prior work. Given the above, I will not recommend the paper be published.

The technical contribution of the paper may be sound and novel, but it appears that it wasn't clear or accessible enough for most of the reviewers. Thus in revising the paper, it may simply be a case of re-writing to improve clarity and the positioning of the paper in the literature. For example, Theorem 3.4 is hard to parse, given the large number of new constant definitions, where the parameter of most interest the mini-batch size (m) is hidden away in several new constants, making it hard to appreciate the speed-up offered by increasing the mini-batch size.

Some further references on SGD + minibatch analysis (though in the smooth case) include [1] in the non-convex setting and [2] in the convex setting. On understanding SGD with momentum in the nonconvex setting, recently (and arguably concurrent work with yours) SGD with momentum was analysed in [3].

[1] Better Theory for SGD in the Nonconvex World, Ahmed Khaled, Peter Richtárik, 2020

[2]  SGD: General Analysis and Improved Rates, Robert M. Gower, Nicolas Loizou, Xun Qian, Alibek Sailanbayev, Egor Shulgin, Peter Richtárik; ICML 2019.

[3] Aaron Defazio, Momentum via Primal Averaging: Theoretical Insights and Learning Rate Schedules for Non-Convex Optimization, arXiv:2010.00406, 2021

**Consistency Experiment:**

NeurIPS has a long history of experimentation. In 2014, NeurIPS ran an experiment in which 10% of submissions were reviewed by two independent committees to quantify the randomness in the review process. This year, we repeated a variant of this experiment to see how the quality of the review process has changed over time.  This paper was part of the experiment and was therefore assigned to two committees (consisting of reviewers, an Area Chair, and a Senior Area Chair) that reached independent decisions.  If both committees made the same recommendation, this recommendation was followed. If a single committee recommended acceptance, the paper was accepted (with the exception of a few cases in which the other committee identified what we considered a fatal flaw, e.g., an error in a key result).

This copy’s committee reached the following decision: **Reject**

The other committee assigned to the paper recommended **Accept (Poster)**.  You can find the other set of reviews, along with any follow up discussion with the authors here:
https://openreview.net/forum?id=PqkKlKQuGZw